# Hyperuricemia and Progression of Chronic Kidney Disease: A Review from Physiology and Pathogenesis to the Role of Urate-Lowering Therapy

**DOI:** 10.3390/diagnostics11091674

**Published:** 2021-09-13

**Authors:** Tao Han Lee, Jia-Jin Chen, Chao-Yi Wu, Chih-Wei Yang, Huang-Yu Yang

**Affiliations:** 1Department of Nephrology, Chang Gung Memorial Hospital, College of Medicine, Chang Gung University, Taoyuan 33305, Taiwan; kate0327@hotmail.com (T.H.L.); Raymond110234@hotmail.com (J.-J.C.); cwyang00@gmail.com (C.-W.Y.); 2Department of Pediatrics, Division of Allergy, Asthma, and Rheumatology, Chang Gung Memorial Hospital, College of Medicine, Chang Gung University, Taoyuan 33305, Taiwan; joywucgu@cgmh.org.tw; 3Department of Health Policy and Management, Johns Hopkins Bloomberg School of Public Health, Baltimore, MD 21205, USA

**Keywords:** hyperuricemia, gout, chronic kidney disease, xanthine oxidase inhibitor, allopurinol, febuxostat

## Abstract

The relationship between hyperuricemia, gout, and renal disease has been investigated for several years. From the beginning, kidney disease has been considered a complication of gout; however, the viewpoints changed, claiming that hypertension and elevated uric acid (UA) levels are caused by decreased urate excretion in patients with renal impairment. To date, several examples of evidence support the role of hyperuricemia in cardiovascular or renal diseases. Several mechanisms have been identified that explain the relationship between hyperuricemia and chronic kidney disease, including the crystal effect, renin–angiotensin–aldosterone system activation, nitric oxide synthesis inhibition, and intracellular oxidative stress stimulation, and urate-lowering therapy (ULT) has been proven to reduce renal disease progression in the past few years. In this comprehensive review, the source and physiology of UA are introduced, and the mechanisms that explain the reciprocal relationship between hyperuricemia and kidney disease are reviewed. Lastly, current evidence supporting the use of ULT to postpone renal disease progression in patients with hyperuricemia and gout are summarized.

## 1. Introduction

Hyperuricemia is described as elevated levels of serum uric acid (UA), which is more commonly known as gout, an acute, symptomatic disease where urate crystals deposit in the joint and cause subsequent inflammatory arthritis. Gout, which is defined as “a progressive metabolic disease characterized by symptomatic hyperuricemia and the deposition of monosodium urate crystals in the joints and soft tissues due to an imbalance in UA uptake, synthesis, or excretion” [1], is the most common form of inflammatory arthritis in men older than 40 years. The initial presentation of gout usually involves severe pain focused on a single joint that spontaneously resolves over a few days or weeks [2,3]. After the resolution of acute episodes, the patient enters a symptom-free interval; however, the serum UA level might still be increased during this interval. According to previous reports, hyperuricemia has been linked to other diseases, including hypertension, diabetes mellitus (DM), cardiovascular disease, and chronic kidney disease (CKD). This study will focus more on reviewing the relationship between hyperuricemia and CKD [4,5,6,7].

CKD is defined as abnormalities of the kidney structure or function presented for more than 3 months, according to the Kidney Disease Improving Global Outcome guidelines [8]. Kidney structure abnormalities can be identified through a kidney biopsy and imaging studies, or inferred from markers, including urinary sediment abnormalities or an increased urine protein excretion rate. A decrease in kidney function refers to a decrease in the glomerular filtration rate (GFR). Following the increasing prevalence of CKD worldwide, its prevention and risk factor identification have gained much attention in the past few years. The relationship between renal function impairment and elevated serum UA levels in patients with asymptomatic hyperuricemia or gout has long been a topic of interest for clinicians. In the 1960s, a report revealed that nearly half of gout patients had some evidence of impaired renal function, and almost all gout patients had glomerular, vascular, or tubulointerstitial scarring on autopsy. According to this study, 18% to 30% of patients with gout have been found to die from end-stage renal disease depending on the severity and duration of gout [9]. Following urate-lowering therapy (ULT) and establishing the pathophysiology between elevated serum UA levels and renal disease, abundant clinical trials and experiments have revealed detailed information about the interaction between these two disorders. In this comprehensive review, we introduce the physiology of UA and its associated genetic variation and then review the epidemiology of hyperuricemia and its comorbidity. Subsequently, we summarize the pathogenesis of renal impairment caused by elevated serum UA levels. Lastly, we review the evidence supporting ULT on slowing renal function impairment in patients with hyperuricemia and gout.

## 2. Physiology of UA

### 2.1. Source of UA

Elevated serum UA levels are exacerbated by diets high in fat, protein, or nucleic acid components. Among these products, purine nucleic acid ingestion exhibited the greatest dietary influence on blood UA levels [10,11]. The majority of the endogenous UA is produced through purine metabolism. A healthy male on a purine-free diet usually has a total purine generation of approximately 1200 mg, and a female of approximately 600 mg, which includes endogenous (daily synthesis rates of approximately 300–400 mg) and exogenous purines (usually approximately 300 mg per day influenced by dietary content) [10,12,13]. Although the production and catabolism of purines are relatively constant between 300 mg and 400 mg per day, the relationship between diet and serum UA are more complex than simple purine intake. As mentioned previously, diet content other than purine also influences UA generation. For instance, provisions with a high fructose content, beer, and soft drinks with high-fructose corn syrup influence fructokinase activation, which leads to rapid UA production [14,15,16].

UA is a product of the purine nucleic acid metabolic pathway. Apart from the enzymes involved in purine nucleotide degradation, xanthine oxidase (XO), also known as xanthine oxidoreductase, plays an important role in the biosynthesis of UA. XO is not only present in the plasma but is also widely distributed in various organs including the liver, gut, lung, kidney, heart, and brain, is normally present as an inactive nicotinamide adenine dinucleotide-dependent cytosolic dehydrogenase precursor, and is processed to its active form via oxidation or proteolytic modification [17]. This enzyme is involved in two important steps of UA metabolism that convert hypoxanthine to xanthine and subsequently convert xanthine to UA.

### 2.2. UA Clearance

In most mammalian species, UA is not the end product of the purine nucleic acid metabolic pathway. UA is further metabolized to the more soluble allantoin by uricase and readily excreted in the urine. However, humans lack the functional uricase enzymes, making UA the final degradation product of the metabolic pathway [18]. In humans, approximately two-thirds of urate is excreted in the urine with normal uricosuria levels of approximately 620 mg/day in adults, with the remainder largely excreted via the gastrointestinal tract [19]. In the kidneys, nearly all of the serum urate is filtered by the glomeruli, but approximately 91–95% is reabsorbed in the proximal tubule via renal tubule transporters, such as urate anion transporter 1 (URAT1, encoded by the SLC22A12 gene) [20,21]. Eventually, approximately 3–10% of the filtered urate appears in the urine after the glomerular filtration, proximal tubule reabsorption, secretion, and post-secretory reabsorption. Thus, hyperuricemia may also lead to hyperuricosuria, defined as a 24 h urinary excretion of over 800 mg in men and 750 mg in women. UA within the urine tends to crystallize when the urine pH is low; therefore, hyperuricosuria combined with acidic urine and low urine volume might result in urate stone formation. Previous reports have pointed out that DM patients had significantly higher urine UA levels and a lower urine pH compared to the nondiabetic population, making DM patients prone to develop urate stones [22,23]. Not only the renal transport system influenced the urate metabolites, but the decreasing of gastrointestinal urate excretion also increased the risk of hyperuricemia. For example, the ABCG2, a high-capacity urate exporter, dysfunction had been reported to increase the risk of hyperuricemia due to a decrease in intestinal urate excretion [24].

### 2.3. Genetic Variations Influence the Serum UA Level

Hyperuricemia should be considered as a combined result of UA overproduction and underexcretion by the kidneys; therefore, all of the genetic variations involved in UA metabolism and elimination could contribute to hyperuricemia.

Several single-locus mutations are known to cause hyperuricemia via UA overproduction. Hypoxanthine-guanine phosphoribosyltransferase (HPRT) is an enzyme that recycles hypoxanthine and guanine. HPRT malfunction results in excessive UA production. Lesch–Nyhan syndrome, an X-linked disorder of HPRT deficiency, is characterized by UA overproduction resulting in precocious hyperuricemia, UA nephrolithiasis, spasticity, and mental retardation [25]. Phosphoribosylpyrophosphate (PRPP) synthetase takes part in the very first step in the de novo synthesis of purines and increasing the activity of PRPP synthetase leads to a purine overproduction and subsequent serum UA elevation. PRPP synthetase overactivity and mutation of the PRPP synthetase 1 gene on Xq22-24 result in hyperuricemia, hyperuricosuria, dysmorphic deafness, and severe mental retardation [26,27]. Treatment with allopurinol, which reduces UA production by inhibiting XO, shows promising effects in controlling hyperuricemia, UA nephrolithiasis, and the metabolic complications of these two disorders.

As mentioned in the previous paragraph, urate was mainly excreted by the kidney and gastrointestinal tract in humans, so hyperuricemia can result from renal and gut underexcretions of UA [19]. Thus, it is not surprising that many novel genes associated with hyperuricemia, identified by genome-wide association studies (GWAS), are involved in the renal urate transport system [28]. URAT1; glucose transporter type 9 (GLUT9); organic anion transporters 1, 3, and 4 (OAT1, OAT3, and OAT4); multidrug resistance-associated protein 4; sodium-coupled monocarboxylate transporter 1 and 2 (SLC5A8, SLC5A12), adenosine triphosphate-binding cassette subfamily G member 2; and PDZ domain-containing 1 (PDZK1) have been identified to cause a variance in serum urate levels according to the GWAS in the past few years [28,29,30,31].

In addition to the proteins directly involved in UA metabolism and excretion, the mechanisms indirectly influencing urate excretion might also affect serum UA levels. Autosomal dominant tubulointerstitial kidney disease (ADTKD), also called medullary cystic kidney disease or familial juvenile hyperuricemic nephropathy, is a group of disorders characterized by precocious hyperuricemia, hyperuricosuria, and progressive CKD. Among these disorders, ADTKD caused by mutation in the uromodulin gene (UMOD gene, encoded in chromosome 16p12) is the most common form of ADTKD and is also known to be involved in its pathogenesis. UMOD is produced exclusively in the thick ascending limbs (TAL) of the loop of Henle, which is involved in intracellular trafficking of the furosemide-sensitive Na-K-2Cl cotransporter and the renal outer medullary potassium channel. Mutations in the UMOD gene cause decreased UMOD production and subsequently decrease the expression of the furosemide-sensitive Na-K-2Cl cotransporter in the apical membrane of the TAL. Decreasing furosemide-sensitive Na-K-2Cl cotransporter expression induces mild natriuresis and volume contraction. To compensate for these changes, the proximal tubule increases sodium reabsorption, which in turn increases proximal urate reabsorption following sodium absorption [32,33,34].

By identifying the genetic variants of hyperuricemia and gout and investigating its pathogenesis, the prediction of treatment responsiveness, for example, ULT, in hyperuricemia and gout is not impossible, making personalized and precision medication more feasible in the near future.

## 3. Hyperuricemia and Renal Disease

According to cross-sectional case-control studies, hyperuricemia and gout have been reported to be comorbid with multiple diseases, including coronary artery disease, hypertension, type 2 DM, or CKD, and are independent risk factors for premature death [35,36,37]. Gout is associated with all the components of the metabolic syndrome, and the prevalence of these comorbidities increases with the gout duration [38]. However, it is still unclear whether elevated serum UA levels are a causal factor in the development of these comorbidities, or whether hyperuricemia and gout share similar risk factors with these disorders.

As mentioned previously, the relationship between hyperuricemia and renal impairment was investigated as early as the 1960s via a postmortem analysis. At the time, the presence of kidney disease in gout patients was considered to be one of the complications of gout and was attributed to the presence of urate crystals in the kidney, leading to the extensive use of the term gouty nephropathy. However, in the 1980s, the viewpoint of the relationship between gout and renal impairment led us to propose that both of these diseases were the consequence of hypertension, causing an initial decline in the renal function and the subsequent elevation of serum UA levels due to decreased urate excretion via the kidneys. Based on this theory, the treatment of hyperuricemia in patients with kidney disease was only recommended to manage gout, renal stones, or, in rare cases, the tumor lysis syndrome. The treatment of asymptomatic hyperuricemia was not recommended at that time.

However, the investigators again redefined the relationship between hyperuricemia and renal impairment during the millennium. After reviewing epidemiological studies, Johnson et al. suggested reconsidering the possibility that hyperuricemia plays a role in directly contributing to cardiovascular or renal diseases. These studies suggest that chronic hyperuricemia is strongly associated with chronic tubulointerstitial disease and a consequent decrease in renal function [39,40,41]. In 2019, the KoreaN cohort Study for Outcomes in patients With Chronic Kidney Disease (KNOW-CKD), a prospective cohort study that enrolled 2042 patients with CKD, tried to investigate the relationship between the level of serum UA and the prevalence of CKD. This study classified patients into quartiles on the basis of their serum uric acid level and had revealed that the prevalence of advanced CKD was higher in patients with a high uric acid level [42]. The association between hyperuricemia and chronic kidney disease was also revealed by a nationwide, cross-sectional cohort study using data from the Japan Chronic Kidney Disease Database. Sofue and his colleagues had reported that the prevalence of hyperuricemia was associated with CKD stages 3b, 4, and 5 by using logistic regression analysis [43]. The animal models also supported this theory by demonstrating that hyperuricemia induced by a fructose-enriched diet, inhibiting uricase or deleting GLUT9 transporters, can result in vascular disease, metabolic syndrome, and glomerular and renal atherosclerosis [44,45]. Most of these diseases can be prevented by early administration of ULT [45,46,47,48]. Further studies suggested that intermittent uricosuria might result in tubular injury by both the direct toxic effects of urate crystals in tubules and the phenotypic transition of tubular cells induced by high concentrations of urate [7,11,49]. To date, it is believed that hyperuricemia is not only the result of renal impairment or sharing the risk factors with CKD, but also can be the cause of renal function impairment.

## 4. Mechanism

According to experimental studies focused on the identification of mechanistic pathways, several mechanisms exist that might explain why serum urate or urinary UA might cause CKD. By understanding these important pathophysiological processes, investigators can design clinical trials more specifically and interpret the outcomes of clinical trials more accurately.

### 4.1. Crystalline Effects

The crystalline effect was discovered initially and is the most widely known negative impact of UA. Glomerular ultrafiltration can be enriched with minerals, proteins, or drug metabolites, which tend to supersaturate under volume depletion or specific urine pH levels. Tumor lysis syndrome, heat stress, and rhabdomyolysis can induce an abundant UA release from cells and transiently increase serum UA levels. The acute elevation of serum UA levels subsequently increases urinary urate levels and precipitates urate crystal formation in the renal tubular lumen, especially in acidic urine or dehydration status [50,51]. In addition to these disorders, certain types of purine ingestion also lead to transient increases in serum UA levels with subsequent uricosuria and crystal formation [7,10]. The formation of crystals induced by acute supersaturation states leads to crystal-associated tubular cell injury and renal function impairment.

In contrast to acute events, long-standing moderate supersaturation can promote crystal formation and tubule obstruction over extended periods of time and even lead to tissue remodeling [52,53,54]. Sellmayr and his colleagues had probed the role of crystal-induced inflammation and macrophages in the pathology of progressive CKD via a novel mouse model, which revealed that UA crystal granulomas develop late in chronic UA crystal nephropathy and contribute to CKD progression via trigger M1-like macrophage-related interstitial inflammation and fibrosis [55]. The long-term elevation of urine UA levels constitutes an endogenous danger signal that activates the inflammasome-mediated inflammatory response [56,57,58]. A previous study revealed that soluble UA enhances pyrin domain containing 3 (NALP3) expression, caspase-1 activation, and interleukin-1β and intercellular adhesion molecule-1 levels in the renal proximal tubule cells in a Toll-like receptor 4-dependent manner (Figure 1) [57]. Another study showed that UA induces renal inflammation via the activation of the nuclear factor-κB signaling pathway [58,59,60]. Ryu et al. also reported that the uptake of UA by tubular epithelial cells might induce epithelial-to-mesenchymal transition by decreasing E-cadherin synthesis and increasing E-cadherin degradation [11]. By inducing the inflammatory activation and cell remodeling, elevated urinary UA levels result in chronic renal function decline.

### 4.2. UA and Renin–Angiotensin–Aldosterone System (RAAS)

Mazzali et al. revealed that rats with hyperuricemia developed hypertension after 3 weeks, which can be prevented or improved through the administration of allopurinol or benziodarone. According to the findings of this animal study, it was concluded that hyperuricemia itself can cause hypertension and renal injury via a crystal-independent mechanism, which can lead to hypertension and renal function impairment through the stimulation of the RAAS and inhibition of nitric oxide synthesis [7,46,61,62,63]. The following experiments also supported the theory that UA first activates the prorenin receptors in proximal tubule cells and then stimulates the intrarenal RAAS, increasing renal renin expression and elevating the serum aldosterone and intracellular angiotensin II levels [62,64,65]. This effect is not only revealed in animal models. One randomized, double-blind, placebo-controlled, crossover trial reported that allopurinol can markedly reduce plasma renin activity and decrease the blood pressure level in adults newly diagnosed with hypertension [66]. Talaat and his colleagues revealed the minimal effects on kidney function and blood pressure level in CKD patients who stopped allopurinol therapy but maintained an angiotensin-converting enzyme inhibitor or angiotensin II receptor blocker. However, CKD patients who did not receive RAAS blockers showed marked deterioration in renal function and blood pressure after the cessation of allopurinol therapy. Thus, it was concluded that asymptomatic hyperuricemia has a deleterious effect on the progression of CKD and hypertension, which can be blocked by RAAS blockers [67].

In addition to the RAAS, UA activates other vasoconstrictors, such as endothelin and thromboxane, and suppresses vasodilatory pathways such as NO [41,68]. The result of the use activation vasoconstrictors and suppression of the vasodilators causes systemic and renal vasoconstriction, thereby reducing the renal plasma flow. The persistently elevated intrarenal pressure causes afferent arteriolar hypertrophy, which impairs the renal autoregulation system and weakens the afferent vasoconstrictive response to systemic pressure (Figure 1). The transmission of systemic blood pressure to the glomeruli promotes CKD progression [69]. Rodriguez-Iturbe et al. revealed that persistent hypertension induced the development of low-grade kidney interstitial inflammation involving T cells and macrophages, which continued even after the factors inducing hypertension were initially removed. The innate and adaptive immunity activated by persistent hypertension initially further results in inflammation-induced impairment of the pressure–natriuresis relationship, dysfunction of vascular relaxation, and sympathetic nervous system overactivity [70]. Afferent arteriolar hypertrophy, interstitial inflammation, and impairment of the pressure–natriuresis relationship might explain the phenomenon that hypertension induced by hyperuricemia is initially responsive to ULT but fails to respond to ULT once renal vascular disease, interstitial inflammation, and salt-sensitive hypertension have developed [71].

### 4.3. Intracellular Effect of UA

Although most previous gout studies have mentioned the extracellular effects of UA, mainly the inflammatory response or toxic effects via crystalline deposition in joint spaces or renal tubules, more recent experimental studies have shown that the primary metabolic and renal effects of UA are mediated by intracellular urate [72,73]. Intracellular UA increases intracellular oxidative stress via the stimulation of nicotinamide adenine dinucleotide (NADPH) oxidase, leading to mitochondrial dysfunction. Intracellular oxidative stress then promotes inflammatory cytokine activation and triggers vascular smooth muscle cell proliferation, which further induces a vasoactive response [41,74]. It seems reasonable that the elevation of the extracellular effect of the UA level will increase intracellular UA, but endogenous production of UA, for example, fructose metabolism via XO, could elevate the intracellular UA levels, even when the serum urate levels are within the normal range [75]. This mechanism might explain the synergistic effects of allopurinol and captopril on the blood pressure level in animal models, which cannot be well explained by the RAAS inhibition mechanism alone [76]. Based on these studies, clinicians realized that ULT might provide additional metabolic protection intracellularly beyond the blockade of the RAAS. To date, many studies have used allopurinol or ULT as an add-on therapy to RAAS blockers, and many studies aim to reveal the possible effect of inhibiting cellular UA uptake by uricosuric agents or using antioxidants to eliminate the intracellular oxidative stress induced by urate.

## 5. Treatment and Evidence of Clinical Trials

Several ULT drugs have been approved in the past few years, which can be classified into three main categories: drugs inhibiting UA synthesis (xanthine oxidase inhibitors (XOi), for example, allopurinol, febuxostat, and topiroxostat); drugs increasing UA excretion (uricosurics, for example, benzbromarone and lesinurad); and drugs enabling systemic metabolic hydrolysis of UA (urolytics, for example, rasburicase and pegloticase). Considering that the main mechanism of uricosuric agents is to increase renal urate excretion, these agents might increase the risk of adverse renal events due to the elevated urinary UA levels by increasing the renal clearance of UA [21,77]. Moreover, urolytic agents are only approved for chronic refractory gout with tophaceous deformities or hyperuricemia induced by tumor lysis syndrome and are prescribed through intravenous administration. Clinical studies investigating the role of ULT in CKD prevention have mostly focused on XOis. XOis inhibit XO to reduce the endogenous production of UA and subsequently lower the serum UA levels and can be categorized into two main classes: classical purine analogs (allopurinol) and nonpurine analog compounds (febuxostat and topiroxostat) [63].

### 5.1. Allopurinol

Allopurinol, a structural isomer of hypoxanthine, was the first ULT introduced in 1966 and remains the standard treatment for patients with hyperuricemia and gout. Although allopurinol is the most frequently prescribed ULT, it is a relatively weak XOi that is rapidly metabolized to oxypurinol and renally cleared. No more than 50% of patients can achieve serum UA levels lower than 6 mg/dL under an allopurinol dose of 300 mg/day, according to previous reports [78,79,80]. In addition, allopurinol has been reported to be associated with several adverse reactions, including gastrointestinal effects, rash, Stevens–Johnson syndrome, and allopurinol hypersensitivity syndrome. The incidence of these complications increases in patients with impaired renal function [81,82,83,84].

In 2014, Bose and his colleagues reported a meta-analysis enrolling eight trials with 476 participants, noting that no significant difference was observed in the change in the GFR from the baseline between the allopurinol and control arms (mean difference [MD] 3.1 mL/min/1.73 m^2^, 95% confidence intervals [CI] −0.9 to 7.1 mL/min/1.73 m^2^), while it abrogated the increase in serum creatinine levels from the baseline in three trials (MD −0.4 mg/dL, 95% CI −0.8 to −0.0 mg/dL) [85]. Several clinical trials and meta-analyses have revealed the benefit of ULT in slowing down the GFR decline. In 2015, Kanji et al. published a meta-analysis that enrolled 19 randomized controlled trials carried out for more than 3 months including 992 participants with CKD stage 3–5, showing that patients in the allopurinol group were associated with a modestly better estimated glomerular filtration rate (eGFR) compared to that in the control group (MD 3.2 mL/min/1.73 m^2^, 95% CI 0.16 to 6.2 mL/min/1.73 m^2^) [86]. In a subgroup analysis in 2017, Su et al. revealed that allopurinol significantly slowed the decline in the eGFR but had no significant influence on proteinuria or albuminuria compared to that in the control group [87].

However, two recent, large, placebo-controlled trials (CKD-FIX and PERL) revealed no benefit from ULT with allopurinol in slowing down CKD progression. In the CKD-FIX study, patients with CKD stage 3 or stage 4 with a urinary albumin to creatinine ratio >265 or an eGFR decrease of at least 3.0 mL/min/1.73 m^2^ of the body surface area in the preceding year had been randomly assigned to allopurinol treatment or the placebo groups. This study revealed that ULT with allopurinol did not slow the decline in the eGFR compared with the placebo in CKD patients with a high risk of progression [88]. In the PERL study, there were 267 participants with type 1 DM and the eGFR ranging from 40.0 to 99.9 mL/min/1.73 m^2^ of the body surface area randomly assigned to allopurinol treatment or the placebo groups. The authors concluded that no meaningful clinical benefits of serum urate reduction with allopurinol were observed on kidney outcomes among patients with type 1 DM with early to moderate diabetic kidney disease [89]. These two clinical trials not only showed negative findings on slowing the eGFR decline by allopurinol, but also revealed the possible concerns regarding the adverse effects of allopurinol treatment. Both of these studies revealed a trend of increasing mortality and serious adverse events in patients on allopurinol [88,89,90,91]. Although some investigators have argued that the study population should be optimized by enrolling younger, nonproteinuric CKD patients with a better preserved GFR, these low-risk patients might maintain a stable GFR even after the placebo treatment [90]. In addition, a recent large cohort study revealed that the lower the eGFR, the higher the prevalence of hyperuricemia and gout [92], and patients with a lower GFR had a higher risk of CKD progression, which might have led the group to consider treatment.

### 5.2. Febuxostat

Febuxostat is a selective, nonpurine-type XOi approved by the U.S. Food and Drug Administration in 2009. Febuxostat was reported to be significantly more potent than allopurinol and was more efficacious at doses of 80 to 120 mg to achieve serum urate levels <6 mg/dL compared to an allopurinol dose of 100–300 mg daily [84,93]. Unlike allopurinol, which is mainly excreted in the kidneys, febuxostat clearance is predominantly mediated by hepatic metabolism, which makes it safer to prescribe to patients with impaired renal function [94,95]. According to a meta-analysis published by Ye et al., the tolerability to febuxostat for the treatment of hyperuricemia is similar to that of allopurinol [93].

In 2019, a meta-analysis enrolling 11 clinical trials with 1317 participants revealed that febuxostat not only showed a significant reduction in serum UA levels but was also associated with a higher eGFR (MD 2.36 mL/min/1.73 m^2^; 95% CI −1.62 to 6.33) among CKD stage 3 and stage 4 patients than those in the placebo group [96]. In the same year, a multicenter, prospective, randomized open-label, blinded endpoint study enrolled 1070 elderly hyperuricemia patients (serum UA > 7.0 to ≤9.0 mg/dL) at risk of cerebral, cardiovascular, or renal disease who were randomized to the febuxostat or nonfebuxostat groups. After 36 months of observation, the development of microalbuminuria and progression to overt proteinuria to ≥300 mg/g of creatinine were significantly lower in the febuxostat-treated group. Based on these results, the authors concluded that febuxostat delayed the progression of renal dysfunction [97]. Hsu et al. reported a nationwide database analysis of CKD stage 5 predialysis patients receiving febuxostat or allopurinol. There were 69.57% allopurinol users and 42.01% febuxostat users requiring long-term dialysis (*p* < 0.0001), and the subgroup analysis showed that the renal benefit of febuxostat was consistent across most patient subgroups and/or propensity score-matched cohorts. The adjusted hazard ratio was 0.65 and 0.66 for long-term dialysis and long-term dialysis or death with febuxostat use, respectively. This study revealed that in the CKD stage 5 predialysis patients, febuxostat users had a lower risk of progression to long-term dialysis or death compared to allopurinol users [98]. Chung et al. had published another study investigating the impact of ULT on the progression and recovery of CKD by analyzing 5860 gout patients. This study had revealed that CKD progression was less common in patients exposed to febuxostat compared to patients exposed to allopurinol even though the febuxostat group patients were older and with more comorbidities [99].

Although the hepatic metabolism characteristic made febuxostat seem to be a better choice than allopurinol, especially in patients with renal function impairment, it is difficult for clinicians to not notice the results of the CARES study published in 2018. The CARES investigators reported a multicenter, double-blind, noninferiority trial involving patients with gout and cardiovascular disease who were randomly assigned to receive allopurinol or febuxostat. In 6190 patients, febuxostat was noninferior to allopurinol with respect to the rates of adverse cardiovascular events; however, the all-cause and cardiovascular mortalities were higher with febuxostat than that with allopurinol. Despite this finding, this study had a high discontinuation rate, with a trial regimen discontinuation rate of 56.6% and discontinued follow-up rate of 45.0% [100,101,102]. To investigate the safety concerns regarding all-cause and cardiovascular mortalities, Zhang et al. published a population-based cohort study that enrolled 24,936 febuxostat initiators matched to 74,808 allopurinol initiators. By analyzing the U.S. Medicare claims data, they concluded that no difference was observed in the risk of myocardial infarction, stroke, new-onset heart failure, coronary revascularization, or all-cause mortality between patients initiating febuxostat and allopurinol [103]. Similar results were reported in the population-based cohort studies published by Hsu et al. and Chen et al. [98,104]. In 2020, a meta-analysis of 17 clinical trials reported that no statistical difference was observed in the risk of major adverse cardiovascular events and cardiovascular events between febuxostat and allopurinol treatment groups [105]. A summary of the meta-analysis focused on investigating ULT and renal disease is shown in Table 1. One recently published multicenter, prospective, randomized, noninferiority trial enrolled patients older than 60 years with at least one additional cardiovascular risk. After observation for 1467 days with a low withdrawal rate (6.2% in the febuxostat group and 5.5% in the allopurinol group), it was concluded that febuxostat is noninferior to allopurinol therapy with respect to hospitalization for nonfatal myocardial infarction or biomarker-positive acute coronary syndrome, nonfatal stroke, or cardiovascular death, and its long-term use is not associated with an increased risk of death or serious adverse events compared with allopurinol [104,106,107].

### 5.3. Combination Therapy: Xanthine Oxidase Inhibitor with Urate Reabsorption Inhibitor

Besides the traditional therapeutic agent for hyperuricemia, the specific urate reabsorption inhibitor had been investigated as the new therapeutic option for gout and hyperuricemia patients. After that the URAT1 inhibitors had been proved to show no inferior efficacy in lowing serum urate level comparing to the existing ULT [110,111], several pilot studies had further investigated the renoprotective effect by combining the urate reabsorption inhibitor with xanthine oxidase inhibitors to achieve a more intensively attenuating serum urate level. It is believed to resolve the problem that urate lowering with xanthine oxidase inhibitor therapy alone may not yield a measurable kidney benefit [112].

Stack et al. had revealed that lowering the serum urate by combining febuxostat and verinurad can reduce the serum urate concentration and albuminuria in patients with type 2 DM, albuminuria, and hyperuricemia, and the effect was rapid and the improvement was sustained for 24 weeks [113]. Although there were no clinically meaningful changes in the eGFR or serum creatinine, the change in albuminuria over a short period of time is now an acceptable surrogate outcome to evaluate whether a treatment may result in slowing CKD progression. The SAPPHIRE study, a phase two trial which plans to randomize 725 patients with CKD, hyperuricemia, and UACR 30–5000 mg/g to verinurad + allopurinol, allopurinol, or placebo is still ongoing [114].

## 6. Conclusions

Hyperuricemia can aggravate renal function impairment through several mechanisms, including the direct toxic effect and inflammasome-mediated inflammatory response induced by urate crystallization, activation of the RAAS, and increased intracellular oxidative stress by stimulating NADPH oxidase. ULT has shown its efficacy in slowing the GFR decline in patients with CKD based on the meta-analysis results. Considering the concerns of allopurinol use in patients with renal impairment and severe adverse events, hepatically metabolized febuxostat seems to be a better choice of ULT in patients with renal impairment with noninferior efficacy on lowering the serum urate level, slowing the eGFR decline, and cardiovascular risk. The strategy to combine xanthine oxidase inhibitor with a urate reabsorption inhibitor might also offer a new opportunity on declined CKD progression in patients with hyperuricemia. However, the definite treatment goal of lowering the serum urate level to slow down renal function impairment and prevent cardiovascular risk in patients with hyperuricemia is still ambiguous, and the data from large and long-term trials are expected to identify the goal of serum UA levels and appropriate doses required for ULT.

## Figures and Tables

**Figure 1 diagnostics-11-01674-f001:**
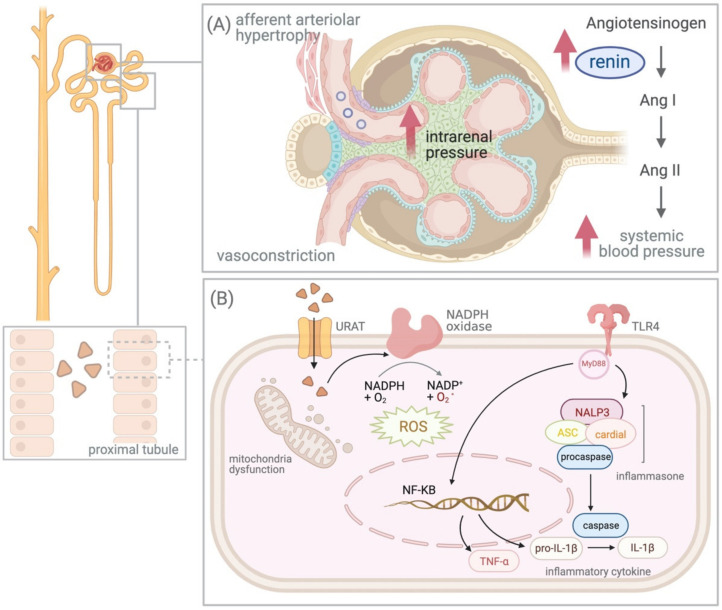
The effects of hyperuricemia on the kidney. (**A**) Stimulation of the renin–angiotensin–aldosterone system and activation of the vasoconstrictors due to hyperuricemia results in elevated intrarenal pressure and afferent arteriolar hypertrophy. (**B**) Soluble uric acid (UA) activates inflammasome pyrin domain containing 3 (NALP3)-induced inflammasome-mediated inflammatory response and intracellular UA increases oxidative stress via the stimulation of nicotinamide adenine dinucleotide (NADPH), leading to mitochondrial dysfunction. Abbreviations: Ang, Angiotensin; URAT, urate anion transporter; ASC, apoptosis-associated speck-like protein containing a caspase recruitment domain; IL, interleukin; NADPH, nicotinamide adenine dinucleotide; NF-κB, nuclear factor-kappa B; TLR, Toll-like receptor; TNF, tumor necrosis factor; ROS, reactive oxygen species.

**Table 1 diagnostics-11-01674-t001:** A Summary of the Meta-Analysis Characteristics that Focused on Urate-Lowering Therapy and Renal Disease.

Study	Year	Sample Size	Population	Intervention Group	Control Group	Outcome				
						eGFR (mL/min/1.73 m^2^)	Creatinine (mg/dL)	Systolic Pressure (mmHg)	Diastolic Pressure (mmHg)	Other Renal-Associated Outcome
Bose et al. [85]	2013	8 RCTs, 476 participants	Patients with normal or mild decreased eGFR	Allopurinol 100–300 mg/day	Placebo, no treatment, or standard therapy	Favor allopurinol (MD: 3.1, CI: −1.2–7.4)	Favor allopurinol (MD: −0.4, CI: 0.0–0.8)	Favor control (MD: −2.7, CI: −7.3–1.9)	Favor control (MD: −1.9, CI: −4.9–1.2)	Allopurinol group ESRD risk (RR 1.01, CI: 0.15–6.98)
Fleeman et al. [108]	2014	4 RCTs, 257 participants	Patients with CKD and hyperuricemia	Allopurinol 100–300 mg/day	Placebo or standard therapy	Favor allopurinol for 6 months (MD: 3.92, CI: −1.33–9.18)	NR	Favor control for 6 months (MD: 1.78, CI: −3.77–7.34) Favor allopurinol in 12 months (MD: −0.32, CI: −5.99–5.35)	Favor allopurinol for 6 months (MD: −0.26, CI: −3.40–2.88) Favor control in 12 months (MD: 0.10, CI: −2.88–3.07)	–
Kanji et al. [86]	2015	19 RCTs, 992 participants	Patients with CKD	Urate-lowering therapy	Placebo or no treatment	Favor allopurinol (MD: 3.2; CI: 0.16–6.2)	Favor allopurinol (MD: 0.629; CI: 0.433–0.825)	Favor allopurinol (MD: 6.6, CI: 2.0–11.1)	Favor allopurinol (MD: 2.1, CI: 0.50–3.7)	–
Pisano et al. [109]	2017	9 RCTs, 695 participants	Patients with CKD and hyperuricemia	Allopurinol, febuxostat, topiroxostat	Placebo or standard therapy	Favor intervention (MD: 2.33, CI: −0.27–4.92)	Favor intervention (MD: −0.05; CI: −0.12–0.02)	NR	NR	Intervention group ESRD risk (RR 0.42, CI: 0.22–0.80)
Lin et al. [96]	2019	11 RCTs, 1317 participants	Patients with CKD and hyperuricemia	Febuxostat 10–80 mg/day	Placebo, allopurinol in two studies benzbromarone in one study	Favor febuxostat (MD: 2.05; CI: −0.24–4.34)	Favor febuxostat (MD: −0.06; CI: −0.21–0.09)	Favor febuxostat (MD: −4.44; CI: −8.08–0.80)	Favor febuxostat (MD: −3.08; CI: −6.25–0.08)	–

Abbreviations: CI, confidence interval; CKD, chronic kidney disease; eGFR, estimated glomerular filtration rate; ESRD, end-stage renal disease; MD, mean difference; NR, not reported; RCTs, randomized controlled trials; XOi, xanthine oxidase inhibitor.

## Data Availability

Not applicable.

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
