# Peer review of "Hyperuricemia and Progression of Chronic Kidney Disease: A Review from Physiology and Pathogenesis to the Role of Urate-Lowering Therapy"

_diagnostics, 2021, doi:10.3390/diagnostics11091674_

Round 1

Reviewer 1 Report

The authors reported a comprehensive review about the impact of hyperuricemia on progression of chronic kidney disease from the view of physiology and pathogenesis to the role of urate lowering therapy. Since this review is well written, I agree that the paper is worthy of being accepted. However, I would ask for a few revisions as follows.

  1. The authors should more describe the epidemiology about the association between hyperuricemia and chronic kidney disease. The authors should cite below recent articles.
    • Sellmayr M, et al. Only Hyperuricemia with Crystalluria, but not Asymptomatic Hyperuricemia, Drives Progression of Chronic Kidney Disease. J Am Soc Nephrol. 2020 Dec;31(12):2773-2792.
    • Sofue T, et al. Prevalences of hyperuricemia and electrolyte abnormalities in patients with chronic kidney disease in Japan: A nationwide, cross-sectional cohort study using data from the Japan Chronic Kidney Disease Database (J-CKD-DB). PLoS One. 2020;15: e0240402.
    • Oh TR, et al. Hyperuricemia has increased the risk of progression of chronic kidney disease: propensity score matching analysis from the KNOW-CKD study. Sci Rep. 2019 Apr 30;9(1):6681.
  2. The Ref. 112 is an abstract of a medical conference of Ref. 111. Therefore, the authors should delete the Ref. 112.
  3. Line 30 to 41, page 1, should be set the same font as the others. 

Author Response

Reviewer 1

The authors reported a comprehensive review about the impact of hyperuricemia on progression of chronic kidney disease from the view of physiology and pathogenesis to the role of urate lowering therapy. Since this review is well written, I agree that the paper is worthy of being accepted. However, I would ask for a few revisions as follows.

Minor concerns

  1. The authors should more describe the epidemiology about the association between hyperuricemia and chronic kidney disease. The authors should cite below recent articles.
    • Sellmayr M, et al. Only Hyperuricemia with Crystalluria, but not Asymptomatic Hyperuricemia, Drives Progression of Chronic Kidney Disease. J Am Soc Nephrol. 2020 Dec;31(12):2773-2792.
    • Sofue T, et al. Prevalences of hyperuricemia and electrolyte abnormalities in patients with chronic kidney disease in Japan: A nationwide, cross-sectional cohort study using data from the Japan Chronic Kidney Disease Database (J-CKD-DB). PLoS One. 2020;15: e0240402.
    • Oh TR, et al. Hyperuricemia has increased the risk of progression of chronic kidney disease: propensity score matching analysis from the KNOW-CKD study. Sci Rep. 2019 Apr 30;9(1):6681.

Reply

Thank you for your valuable comment. We had added more detail to emphasize the association between hyperuricemia and chronic kidney disease and cited the reference as your suggestion. The modified sentence was in page 4, line 185-194, and page 5, line 226-230, in the revised manuscript as follows:

“In 2019, the KoreaN cohort Study for Outcomes in patients With Chronic Kidney Disease (KNOW-CKD), a prospective cohort study that enrolled 2042 patients with CKD, tried to investigate the relationship between the level of serum UA and the prevalence of CKD. This study classified patients into quartiles on the basis of their serum uric acid level and had revealed that the prevalence of advanced CKD was higher in patients with a high uric acid level. The association between hyperuricemia and chronic kidney disease was also revealed by a nationwide, cross-sectional cohort study using data from the Japan Chronic kidney disease database. Sofue and his colleagues had reported that the prevalence of hyperuricemia was associated with CKD stages 3b, 4, 5 by using logistic regression analysis.”

“Sellmayr and colleagues had probed the role of crystal-induced inflammation and macrophages in the pathology of progressive CKD via a novel mouse model, which revealed that UA crystal granulomas develop late in chronic UA crystal nephropathy and contribute to CKD progression via trigger M1-like macrophage-related interstitial inflammation and fibrosis.”

  1. The Ref. 112 is an abstract of a medical conference of Ref. 111. Therefore, the authors should delete the Ref. 112.

Reply

Thank you for your kindly reminder. We had modified the reference as your suggestion.

  1. Line 30 to 41, page 1, should be set the same font as the others.

Reply

Thank you for your kindly reminder. We had set the font of line 30 to 41, page 1 as other paragraphs.

Reviewer 2 Report

  1. The various aspects of the relationship between hyperuricemia and renal disease, including the physiology of uric acid, the reciprocal mechanism between hyperuricemia and kidney disease, and the renal protective effects of urate lowering therapy were well summarized in this article.
  2. Please briefly provide the explanation about ”gut underexcretion of UA” (line 125).
  3. It would be better to remove the subtitle of “3.1. Hyperuricemia and renal disease (line 163)’ and modify the title of “3. Epidemiology of hyperuricemia (line 155)”.

Author Response

Reviewer 2

The various aspects of the relationship between hyperuricemia and renal disease, including the physiology of uric acid, the reciprocal mechanism between hyperuricemia and kidney disease, and the renal protective effects of urate lowering therapy were well summarized in this article.

Minor concerns

  1. Please briefly provide the explanation about ”gut underexcretion of UA” (line 125).

Reply

Thanks for your insightful comment. We had modified our sentence in this paragraph to clarify that the urate excretion might be influenced while the urate transport system was affected. Considering that the renal and gastrointestinal systems were the main pathways to excretion urate from the human body, the underexcretion might increase the risk of hyperuricemia. The modified sentence was in page 3, line 107-110 and line 129-133, in the revised manuscript as follows:

“Not only the renal transport system influenced the urate metabolites, the decreasing of gastrointestinal urate excretion also increased the risk of hyperuricemia. For example, the ABCG2, a high-capacity urate exporter, dysfunction had been reported to increase the risk of hyperuricemia due to decrease intestinal urate excretion.”

“As mentioned in the previous paragraph, urate was mainly excreted by the kidney and gastrointestinal tract in humans, so hyperuricemia can result from renal and gut underexcretion of UA. Thus, it is not surprising that many novel genes associated with hyperuricemia, identified by genome-wide association studies (GWAS), are involved in the renal urate transport system”

  1. It would be better to remove the subtitle of “3.1. Hyperuricemia and renal disease (line 163)’ and modify the title of “3. Epidemiology of hyperuricemia (line 155)”.

Reply

Thanks for your kindly reminder. We had modified the subtitle and title as your suggestion.
